# A Two-Stage Multi-Agent EV Charging Coordination Scheme for Maximizing Grid Performance and Customer Satisfaction

**DOI:** 10.3390/s23062925

**Published:** 2023-03-08

**Authors:** Adil Amin, Anzar Mahmood, Ahsan Raza Khan, Kamran Arshad, Khaled Assaleh, Ahmed Zoha

**Affiliations:** 1Department of Electrical Engineering, Mirpur University of Science & Technology (MUST), Mirpur 10250, Pakistan; 2James Watt School of Engineering, University of Glasgow, Glasgow G12 8QQ, UK; 3Artificial Intelligence Research Center (AIRC), College of Engineering and Information Technology, Ajman University, Ajman P.O. Box 346, United Arab Emirates

**Keywords:** charging cost, electric vehicles, power loss, voltage deviation, waiting time

## Abstract

Advancements in technology and awareness of energy conservation and environmental protection have increased the adoption rate of electric vehicles (EVs). The rapidly increasing adoption of EVs may affect grid operation adversely. However, the increased integration of EVs, if managed appropriately, can positively impact the performance of the electrical network in terms of power losses, voltage deviations and transformer overloads. This paper presents a two-stage multi-agent-based scheme for the coordinated charging scheduling of EVs. The first stage uses particle swarm optimization (PSO) at the distribution network operator (DNO) level to determine the optimal power allocation among the participating EV aggregator agents to minimize power losses and voltage deviations, whereas the second stage at the EV aggregator agents level employs a genetic algorithm (GA) to align the charging activities to achieve customers’ charging satisfaction in terms of minimum charging cost and waiting time. The proposed method is implemented on the IEEE-33 bus network connected with low-voltage nodes. The coordinated charging plan is executed with the time of use (ToU) and real-time pricing (RTP) schemes, considering EVs’ random arrival and departure with two penetration levels. The simulations show promising results in terms of network performance and overall customer charging satisfaction.

## 1. Introduction

Growing concerns regarding climate change, global warming and fossil fuel depletion have motivated the widespread adoption of electric vehicles (EVs). The estimates indicate that EVs will dominate the transportation sector till 2050 [1]. The increase in EV adoption rate is very encouraging; however, the large-scale integration of EVs is challenging the stability and smooth operation of the power supply system due to the increased variations in load demand [2]. The large-scale, uncontrolled penetration of EVs into the electric grid increases the power flow in cables, voltage unbalance, transformer overloading, harmonic problems, etc. [3,4]. Thus, the simultaneous charging of a large number of EVs directly affects the electric grid’s stability. For example, several hundred EVs may request charging from the electric grid over a short span of time. This will cause a sudden increase in load demand and stress the supply network. The EV charging activities require suitable management and control approaches for reliable operation. In the context of a smart grid, the control and automation modules communicate with each other to achieve the set objectives [5]. Communication technologies enable the transfer of information from various parts of the grid. This brings opportunities to implement better operation and control strategies than conventional solutions. In this respect, the communication framework is crucial to support the automated and intelligent management and control functions in electrical power systems. However, communication technologies and network security issues are the main concerns arising due to many electronic meters, sensors, EVs, control and automation devices and distributed generation [6]. EVs share information such as battery state of charge (SoC) and charging requirements along with other important parameters with the EV aggregator. Therefore, appropriate communication protocols for monitoring and managing the charging activities of EVs are important.

Using smart grid communication facilities, EV charging activities can be coordinated, and problems associated with unregulated handling can be minimized while meeting load demand. However, it is challenging for grid operators to manage EV charging activities simultaneously with optimal grid operation. To satisfy the charging requirements of EV users without deteriorating the performance of the electric grid, it is essential to distribute the control of charging activities among the multiple charging agents known as “EV Charging Aggregator Agents”. This approach reduces the search space and provides quality solutions as each aggregator agent will supervise the charging activities of its region only. It will have no concerns with the charging activities in the neighborhood. This way, a hierarchy control is required, which could simultaneously handle the network operation and effectively manage EVs’ charging activities. Furthermore, EV charging is a random process that can take place either at home, at charging stations or even at the workplace. Approximately 50–80% of EV customers prefer overnight charging at home, followed by 15–20% who use workplace charging as a secondary option [7,8]. Previous research work related to EV charging either focused on home-based charging or considered commercial/workplace charging. Generally, the EVs’ charging demand and their impact on the electricity network change regarding location and time [9]. Therefore, it is necessary to investigate the impact of EV charging considering home and workplace platforms with day and night charging schedules to create a useful plan based on customers’ preferences. 

## 2. Related Work

With the increasing introduction of EVs in the transportation sector, it is essential for EV charging aggregators to effectively manage the load demand of EVs by taking into account the needs of both the electric grid and EV users. The EV charging scheduling problem is a non-deterministic complex combinatorial optimization problem in which the computational complexity increases with the penetration of EVs [10]. Several studies on the optimal charging scheduling of electric vehicles have been carried out in the literature, considering various objectives. In [11,12], the power loss is minimized through the coordinated charging operation of EVs at the home level. In [13], the EV charging problem is formulated as a multi-objective problem to minimize the cost and system stress for the residential network. A bi-level EV aggregator-based charging scheme is presented in [14] to achieve the peak shaving and valley filling objectives. The proposed method is demonstrated on a real-world medium voltage distribution network. In [15], the authors presented a weighted sum-based multi-objective plug-in electric vehicle (PEV) charging coordination scheme to minimize power loss and charging costs and maximize the operating capacity of the electric grid. In [16], a coordinated charging framework for EV aggregators is presented to shave the peak demand of the system. In [17], various charging strategies are investigated to coordinate the EV charging operation for the residential network to minimize power loss, power consumption and charging costs. A two-layer smart charging model is proposed by [18] to achieve user satisfaction presented in terms of the number of users whose SoC gain at departure time exceeds 80% of the target power.

In previous works [11,12,13,15,17,19], the single central control unit mainly performed the EV charging process. This method of coordination needs to deal with large solution space as all the EVs operating in the network are supervised by a single control unit. Contrary to this approach, few studies [14,16,18] presented distributed control among multiple EV aggregators for the EV charging coordination process. With this distributed control among EV aggregator agents, the solution space size reduces as each EV aggregator agent manages the EV operating in its locality. These studies, however, only focused on grid performance, and EV customers’ charging satisfaction has not been considered in these studies. Furthermore, the mentioned work only considered the EVs home charging and ignored commercial activities since EV users have the same opportunity to charge their EVs at work, where their EVs remain idle throughout the duration of their stay at the workplace. The comparison of the proposed work with the existing literature is summarized in Table 1.

This work aims to coordinate the charging activities among multiple EV aggregator agents to maximize the grid performance by minimizing network power losses and voltage deviations. The work also focuses on maximizing customers’ satisfaction by minimizing charging costs and EV waiting times. These objectives are achieved in sequential order by implementing a two-stage method. In the first stage, the distribution network operator (DNO) receives the charging request from each EV aggregator contributing to the scheduling process. The DNO then checks the maximum demand constraint and distributes power among the participating aggregator agents to achieve minimum power losses and voltage deviations. Once the suitable power is allocated to the EV aggregator agents, their algorithms are executed to ensure the optimal charging operation at the minimum cost and waiting time. The presented method is tested and verified on an IEEE-33 bus medium voltage network coupled with a low-voltage network.

The major contributions of this work can be summarized as follows:A two-stage coordination framework for EV charging and scheduling is presented to maximize network performance and customer charging satisfaction;Various indices for power loss, voltage profile, charging cost and waiting time are devised for the coordinated charging operation of EVs, and the customers’ charging satisfaction is investigated with the time of use (ToU) and real-time pricing (RTP);The coordinated framework among muti EV aggregator agents is established to manage the charging activities at residential and workplace platforms;The proposed approach bears a low computational burden.

The rest of the paper is structured as follows. Section 3 explains the aggregator-based EV charging management (AECM) architecture. In Section 4, a two-stage EV charging management strategy is presented. The test cases and simulation setup are given in Section 5. The results and relevant discussion are provided in Section 6. A comparison of the proposed work with the reported studies is given in Section 7. Finally, the conclusions are drawn in Section 8.

## 3. AECM Architecture 

This section presents an AECM architecture to control the charging activities of EV customers. The AECM provides the coordination between EVs and DNO to realize the optimal operation of the distribution network and the charging satisfaction of EV customers. The designed model is implemented in two stages and shown in Figure 1. 

The first stage is administered by the DNO, which is responsible for the optimal operation of the distribution network. The EV charging activities at the residential and commercial platforms are supervised by the respective aggregators. The aggregators collect the charging requests from the EV customers and communicate with the DNO to allocate charging power. The DNO then executes its algorithm and distributes the charging power among the participating aggregators. Subsequently, the aggregators manage the charging operation of EVs by considering their preferences. 

## 4. Two-Stage EV Charging Management Strategy

### 4.1. Scenario Design 

An EV has various charging facilities to charge its battery. These facilities include residential, workplace, commercial and public charging supply points. Each charging supply point has a specific charging level. For example, home-based and workplace charging facilities operate at level-I or level-II. They are classified based on electrical parameters, such as current, voltage and power rating. The level-I charging standard employs single-phase 115 VAC/15 A or 230 VAC/6 A and can supply 1.5 kW of power to the EV battery [20]. The 230 VAC/30 A and 7 kW power rating standards are part of level-II charging and are commonly used at home and public charging stations [20]. The level-III charging mode is a comparatively fast charging solution and is the most suitable for commercial vehicles [20]. In many countries, private EVs dominate the transport sector, and approximately 50–80% of EV customers prefer overnight charging at home, followed by 15–20% of EV customers who use workplace charging as a secondary option [7,8]. Moreover, private EVs have short travel times and long idle or parking times [19]. Based on these facts, we have developed a charging scenario for private EVs and managed their day and night charging activities via EV aggregators using workplace and residential charging platforms with level-II charging infrastructure. Due to the uncertainty of the arrival and departure of EVs and their increasing penetration into the power grid, it is difficult for the grid operator to maintain grid operational efficiency and achieve the load satisfaction of every individual EV customer in a shorter lead time. It is essential in real-world problems to achieve the desired objectives with reasonable computational effort [21]. If the problem is divided into two stages instead of solved as one unit, we can achieve our objectives and shorten the algorithm execution time. In the first stage, the DNO receives the charging load demand from the EV aggregators and allocates the appropriate power to each participating EV aggregator while minimizing grid power losses and voltage deviations. In contrast, the charging satisfaction of individual EV customers is ensured in the second stage of the proposed scheme. 

For every EV participating in the charging event, arrival time, departure time, arrival SoC and desired departure SoC are considered with minimal cost and delay. 

In this work, a two-stage coordination mechanism between DNO and EV aggregators is established to optimize the operation of the distribution network as well as EV charging events. The stages of the proposed model are described in the following subsections.

### 4.2. Stage-1: Power Distribution to the Aggregators 

The first stage is implemented at the level of DNO, which is responsible for the optimal operation of the electrical grid. At this point, EV aggregators collect information such as arrival time, departure time, arrival SoC and desired departure SoC from EVs operating at their locations. Afterward, the aggregators send their charging load request to the DNO. On receiving this request, the DNO executes its algorithm and distributes the available power among EV aggregators to achieve minimal power losses and voltage deviations. The minimization function, similar to [13], is given by Equation (1) and is used to distribute the available power among the EV aggregators optimally.
(1)$$\min\left(f_{1}\right)=\min\left(PLI+VDI\right)$$
where PLI is the power loss index and VDI is the voltage deviation index, and they are calculated as Equations (2) and (3), respectively.
(2)$$PLI=\left|1-\sum_{\Delta t=1}^{T}\left(\frac{PL_{\Delta t}^{w/cp}}{PL_{\Delta t}^{wcp}}\right)\right|\begin{array}{cc} & \forall \Delta t\in T \end{array}$$
where $PL_{\Delta t}^{wcp}$ and $PL_{\Delta t}^{w/cp}$ are the power losses with and without charging power allocation to the EV aggregators, respectively, and can be calculated as follows:(3)$$PL^{wcp},PL^{w/cp}=\sum_{l=1}^{L}\left(I_{\left(l,\Delta t\right)}^{2}\times R_{\left(l,\Delta t\right)}\right)\begin{array}{cc} & \forall l\in L \end{array},\forall \Delta t\in T$$

$I_{\left(l,\Delta t\right)}$ and $R_{\left(l,\Delta t\right)}$ are the current and resistance of the *l*th branch of the network having *L* number of branches. The *VDI* in Equation (1) represents the voltage deviation index which can be calculated in Equation (4).
(4)$$VDI=\sum_{\Delta t=1}^{T}\frac{{\left(\left|V_{}^{ref}-V_{\Delta t}^{\min}\right|\right)}^{2}}{V_{}^{ref}}\begin{array}{cc} & \forall \end{array}\Delta t\in T$$
where $V_{}^{ref}$ is the reference voltage and assumed as 1.0 p.u. The minimum voltage recorded at any bus in a time Δt is represented by $V_{\Delta t}^{\min}$.

#### Constraints 

The optimal solution must not violate the following set of constraints. 

Voltage Limits:

The voltage at each bus *b* from the set of *B* number of buses must be within the defined limits.
(5)$$V_{b}^{\min}\leq V_{\left(b,\Delta t\right)}\leq V_{b}^{\max}\begin{array}{cc} & \forall b\in B \end{array},\forall \Delta t\in T$$
where $V_{b}^{\min}$ and $V_{b}^{\max}$ in Equation (5) are the minimum (0.90 p.u.) and maximum (1.10 p.u.) voltage limits for the bus b, respectively, and denote the voltage in p.u. on the bus *b* for a time slot Δt. 

Power Allocation and Distribution Constraints

We have introduced power allocation and distribution constraints to ensure that the power is not allocated to nonparticipant EV aggregators and that the charging demand of critical and high-priority EV customers is consistently met. For this purpose, Equations (6) and (7) should not be violated during the optimization process.
(6)$$P_{\left(a,\Delta t\right)}^{}=0\begin{array}{cc} & \forall a\notin A, \end{array}\forall \Delta t\in T$$
(7)$$P_{\left(a,\Delta t\right)}^{\min}\leq P_{\left(a,\Delta t\right)}^{}\leq P_{\left(a,\Delta t\right)}^{\max}\begin{array}{cc} & \forall a\in A, \end{array}\forall \Delta t\in T$$

$P_{\left(a,\Delta t\right)}^{}$ represents the power allocation to the candidate aggregator *a* from the set of aggregators *A* and $P_{\left(a,\Delta t\right)}^{\min}$ and $P_{\left(a,\Delta t\right)}^{\max}$ which are the minimum and maximum power distributions among the participating EV aggregators per their charging needs. 

Instantaneous Maximum Demand Constraint

To ensure that the instantaneous load demand of both residential and commercial feeders is within their rated capacity, the maximum demand limit constraints are considered and represented in Equations (8)–(10).
(8)$$\sum_{s=1}^{S}RD_{\left(s,\Delta t\right)}\leq RD_{\max}\begin{array}{cc} & \forall s\in S,\Delta t\in T \end{array}$$
(9)$$\sum_{c=1}^{C}CD_{\left(c,\Delta t\right)}\leq CD_{\max}\begin{array}{cc} & \forall c\in C,\Delta t\in T \end{array}$$
(10)$$\left(\sum_{s=1}^{S}RD_{\left(s,\Delta t\right)}+\sum_{c=1}^{C}CD_{\left(c,\Delta t\right)}\right)\leq GD_{\max}\begin{array}{cc} & \forall s\in S,c\in C,\Delta t\in T \end{array}$$
where $RD_{\left(s,\Delta t\right)}$ and $CD_{\left(c,\Delta t\right)}$ are the instantaneous load demand at residential and commercial feeders, respectively. $RD_{\max}$, $CD_{\max}$ and $GD_{\max}$ represent the maximum demand at the residential, commercial and grid levels, respectively.

Stage 1, presented above, determines the optimal distribution of charging power among the participating EV aggregators while maintaining the stated constraints.

### 4.3. Stage-2: Coordinated EV Charging 

Once each participating EV aggregator agent receives the charging power from the DNO, its role is to manage the charging activities so that the EV customers operating in its vicinity can enjoy economical charging without any delay. To this end, two satisfaction indices known as (i) Minimum Cost Index and (ii) Minimum Waiting Time Index have been introduced to ensure the charging requirements of EV customers. Thus, the goal of an EV aggregator agent is to achieve the charging satisfaction of EV customers in terms of cost and time. The objective function of each participating agent is given by Equation (11).
(11)$$\min\left(f_{2}\right)=\min\left(CCI+WTI\right)$$

It consists of two normalized indices: (i) normalized EV charging cost, Equation (12), and (ii) normalized waiting time, Equation (13).
(12)$$CCI=\frac{\sum_{\Delta t=\Omega_{s}}^{\Omega_{e}}\sum_{a=1}^{A}\sum_{e=1}^{Nev}\left(\Phi_{\left(e,a,\Delta t\right)}^{}\times \Delta t\right)\times \rho_{\Delta t}}{\gamma \times \lambda }\begin{array}{cc} & \forall e\in Nev, \end{array}a\in A,\left(\Delta t,\Omega_{s},\Omega_{e}\right)\in T$$
where $\Phi_{\left(e,a,\Delta t\right)}^{}$ denotes the charging power of *e*th EV from the set of *Nev* EVs selected through the optimization process for the charging interval $\Omega_{s}$ to $\Omega_{e}$. The ‘*a*’ represents the *a*th EV aggregator agent from the set of *A* number of agents. The charging cost in a time step Δt is given by $\rho_{\Delta t}$ and the total time slots are given by λ. We have introduced γ as the incentive for the contribution towards CO_2_ emission reduction, which is assumed as USD 100.
(13)$$WTI=\sum_{a=1}^{A}\sum_{e=1}^{Nev}\frac{\left|\Upsilon_{\left(a,e\right)}^{ari}-\Upsilon_{\left(a,e\right)}^{con}\right|}{\Upsilon_{\left(a,e\right)}^{sty}}\begin{array}{cc} & \forall a\in A,e\in Nev,\left(\Upsilon_{\left(a,e\right)}^{ari},\Upsilon_{\left(a,e\right)}^{con},\Upsilon_{\left(a,e\right)}^{sty}\right) \end{array}\in T$$
(14)$$\Upsilon_{\left(a,e\right)}^{sty}=\left|\Upsilon_{\left(a,e\right)}^{ari}-\Upsilon_{\left(a,e\right)}^{dep}\right|$$
where $\Upsilon_{\left(a,e\right)}^{ari}$ is the arrival time of the *e*th EV of the *a*th agent. $\Upsilon_{\left(a,e\right)}^{con}$ denotes connection time *e*th EV, belongs to the *a*th agent, and it is determined through the optimization process so that Equation (13) can be minimized. The total stay time of the *e*th EV controlled by the *a*th agent is represented by $\Upsilon_{\left(a,e\right)}^{sty}$ and calculated by Equation (14). 

#### Constraints 

To ensure that EVs are charged at a low cost and with less waiting time, the following constraints are considered.

Stay Time Constraint

To meet the changing requirements of EVs, the stay time of each EV must be greater than or equal to the time required to achieve the requested SoC.
(15)$$\Upsilon_{\left(a,e\right)}^{sty}\geq \Upsilon_{\left(a,e\right)}^{req\_soc}$$

The time required to achieve the requested SoC, $\Upsilon_{\left(a,e\right)}^{req\_soc}$, can be calculated by Equation (16) where $\psi_{\left(e,a\right)}$ and $\Phi_{\left(e,a\right)}$ are the battery capacity (kwh) and charger rating (kW) with the $\eta_{\left(e,a\right)}$ efficiency of the *e*th EV managed by the *a*th agent. $SoC_{\left(e,a\right)}^{req}$ and $SoC_{\left(e,a\right)}^{ari}$ are the requested and initial SoC levels, respectively, of the eth EV’s battery.
(16)$$\Upsilon_{\left(a,e\right)}^{req\_soc}=\frac{\psi_{\left(e,a\right)}\times \left(SoC_{\left(e,a\right)}^{req}-SoC_{\left(e,a\right)}^{ari}\right)}{\Phi_{\left(e,a\right)}^{}\times \eta_{\left(e,a\right)}}$$

The optimization problem described in stage-2 is solved by using a genetic algorithm (GA). The proposed two-stage research design along with the interaction stages and the control action between the two stages are explained in the next section. 

### 4.4. Proposed Research Design with Information Exchange and Control Process 

The proposed two-stage research design is summarized in Figure 2.

#### 4.4.1. Steps for Stage-1 

Stage-1 involves the following step:i.Input Data

Stage-1 uses the grid topology as the input data, which include line and load data from [11]. Besides this, 24 h load profiles for commercial and residential areas are also used at the input stage. These load profiles are described in Section 5.1.

ii.Decision Variables

The stage-1 uses charging power allocation $P_{\left(a,\Delta t\right)}^{}$ as the decision variable. It is the amount of power that should be distributed among the EV aggregator agents to optimize the objective functions of stage 1. 

iii.Objective Function

In order to maintain the performance of the network, stage-1 considers the power loss index PLI and the voltage deviation index VDI as the optimization objectives. For each time when power is allocated to the aggregators, these optimization objectives are evaluated.

iv.Optimization Method

PSO is used as the optimization method of stage-1, which determines the value of $P_{\left(a,\Delta t\right)}^{}$ so that objective functions are minimized. This algorithm works every time when the aggregator requests to allocate the charging power. 

v.Output data

The output of stage-1 is derived from objective functions at each execution of the PSO algorithm. We have illustrated the output as the power loss and voltage deviation in Section 6. 

#### 4.4.2. Steps for Stage-2 

Stage-2 involves similar steps with detail as below:i.Input Data

Stage-2 uses the EV charging data such as EV penetration level, charger rating and the SoC of EV batteries. The relevant information can be obtained from Section 5.2. Besides this, stage-2 also uses the EV mobility data and electricity pricing signals given in Section 5.3 and Section 5.4, respectively.

ii.Decision Variables

Once the power is allocated to each aggregator, each aggregator considers charging power $\Phi_{\left(e,a\right)}$ and connection time $\Upsilon_{\left(a,e\right)}^{con}$ as the decision variables. These variables define the candidates EVs involve in the charging process.

iii.Objective Function

To satisfy the distinct charging requirements of every individual EV customer, we have considered charging cost and waiting time minimization as the objectives of stage-2. These objectives are evaluated during each charging interval.

iv.Optimization Method

GA is used as the optimization method of stage-2, which determines the charging power of the best candidate EVs and their connection times so that the stated objective functions are minimized. The algorithm runs every time after power is allocated to the aggregator from stage-1, and it is to be distributed among the EVs in queue.

v.Output Data

The output of stage-2 is obtained in terms of objective functions for each execution of the GA algorithm. We have illustrated the output as the connecting time of EVs and their total charging cost in Section 6. 

The interaction between the two stages and the control action is shown in Figure 3 and explained below. 

The execution of each strategy is elaborated in the following steps:

**Step 1:** The DNO records the residential $\sum_{s=1}^{S}RD_{\left(s,\Delta t\right)}$ and commercial $\sum_{c=1}^{C}CD_{\left(c,\Delta t\right)}$ load profiles for each time interval Δt.**Step 2:** The EV aggregator agent receives the charging information from the EVs operating in its territory. The information includes: arrival time $\Upsilon_{\left(a,e\right)}^{ari}$, departure time $\Upsilon_{\left(a,e\right)}^{dep}$, SoC at the arrival $SoC_{\left(e,a\right)}^{ari}$, requested SoC at departure $SoC_{\left(e,a\right)}^{dep}$ and customer charging preference/priority. Each participating EV aggregator agent requests the DNO for the allocation of charging power. **Step 3:** The DNO checks the condition of the maximum demand limit for both residential and commercial feeders, i.e., $RD_{\max}$ and $CD_{\max}$. Once the condition is met, the DNO executes the objective function (*f_1_*) and allocates the charging power $P_{\left(a,\Delta t\right)}^{}$ to each participating agent.**Step 4:** The participating agents receive the allocated power to fulfil the charging demand of EVs.**Step 5:** The EV aggregator agents execute the objective function (*f_2_*) by considering customers’ preference/priorities.**Step 6:** The EV aggregator agents then send the charging signal to the best selected EVs to start charging.**Step 7:** Once the charging process starts, the EV aggregator agents compute the charging load and update the DNO about it.**Step 8:** After receiving the information about the connected load of EVs, the DNO makes sure that the connected charging load does not have any effect on the network performance.

The pseudo-code of the proposed scheme for both stages is given in Algorithms 1 and 2 below.

**Algorithm 1.** Power Allocation to EV Aggregator Agents **Input:**$\sum_{s=1}^{S}RD_{\left(s,\Delta t\right)}$$,\;RD_{\max}$$,\sum_{c=1}^{C}CD_{\left(c,\Delta t\right)}$$,CD_{\max}$,e∈Nev, a∈A$,\;SoC_{\left(e,a\right)}^{ari}$$,SoC_{\left(e,a\right)}^{req}$$,\Upsilon_{\left(a,e\right)}^{ari}$$\Upsilon_{\left(a,e\right)}^{dep}$$,\Phi_{\left(e,a\right)}$, Δt∈T,
**Output:**

$P_{\left(a,\Delta t\right)}$


Power allocated to ath aggregator in time stepΔt


Each EV Aggregator Agent a from the set of A agents collects $SoC_{\left(e,a\right)}^{ari}$, $SoC_{\left(e,a\right)}^{req}$, $\Upsilon_{\left(a,e\right)}^{ari}$, $\Upsilon_{\left(a,e\right)}^{dep}$ and $\Phi_{\left(e,a\right)}$ time step Δt
EV Aggregator Agent a calculates, stay time $\Upsilon_{\left(a,e\right)}^{sty}$ and time required to achieve desired SoC $\Upsilon_{\left(a,e\right)}^{req\_soc}$ of *a*th EV by using Equations (14) and (16). Respectively.**If**  $\left(\sum_{s=1}^{S}RD_{\left(s,\Delta t\right)}\leq RD_{\max}\right)$
**|**
$\left(\sum_{c=1}^{C}CD_{\left(c,\Delta t\right)}\leq CD_{\max}\right)$
**then**DNO executes the PSO algorithm as follows.
i.**Initialize PSO parameters:** [Max. Iteration, population size, upper & lower bound of population, velocity limits].ii.**Randomize the population:** [set of power allocation to the aggregator agents].iii.**For each solution**iv.Evaluate the fitness functions: [using Equations (2) and (4)] and verify the constraints [using Equations (5)–(7) and (10)].v.Update local best *(pBest)* [individual best power allocated]vi.Update global best *(gBest)* [overall best power allocated]vii.**Do**viii.**For each solution**ix.Update solution velocity and position {search for new solutions}x.Evaluate the fitness functions: [using Equations (2) and (4)] and verify the constraints [using Equations (5)–(7) and (10)].xi.Update local best *(pBest)* [individual best power allocated]xii.Update global best *(gBest)* [overall best power allocated]xiii.**While** (not reached to maximum iteration)

**End process**


**Algorithm 2:** Power Allocation to EV Aggregator Agents **Input:**e∈Nev, a∈A$,\;P_{\left(a,\Delta t\right)}$$,\Upsilon_{\left(a,e\right)}^{ari}$$,\;\Upsilon_{\left(a,e\right)}^{dep}$$,\;SoC_{\left(e,a\right)}^{ari}$$,SoC_{\left(e,a\right)}^{req}$$,\Phi_{\left(e,a\right)}$, Δt∈T, ToU, RTP price signal. **Output:**$\;\mathrm{Optimal}\;\mathrm{charging}\;\mathrm{plan}\;i.e.,\;\mathrm{charging}\;\mathrm{power}\;\Phi_{\left(e,a\right)}$$\mathrm{of}\;\mathrm{candidate}\;\mathrm{EVs}\;\mathrm{and}\;\mathrm{their}\;\mathrm{connection}\;\mathrm{time}\;\Upsilon_{\left(a,e\right)}^{con}$ considering their preferences.
Each EV Aggregator Agent a from the set of A agents receive $P_{\left(a,\Delta t\right)}$ Power from DNO to distribute it among EVs of its locality.EV Aggregator Agent a then sort EVs based on their preferences i.e., early charging and/or low charging cost.EV aggregator agents execute GA to determine the best EVs among EVs available in queue and decide their connection time as follows.
i.**Initialize GA parameters:** [Max. Iteration, No. Xoms, probability of crossover, probability of mutation, it = current iteration, pop(it) = current population].ii.No. Xoms(it) = **Randomize the initial generation of chromosome:** [charging event of randomly selected EVs].iii.**While** it < Max. Iteration **do**iv.rand_c_, rand_m1_, rand_m2_ = random no. ranges from 0 and 1.v.chrm1, chrm2 = select 2 chromosomes from pop (it)vi.eliminate chrm1, chrm2 from pop (it)vii.**if** rand_c_
**<** probability of crossover, **then**viii.chrm3, chrm4 = crossover chrm1 & chrm2 [c2 offspring are generated]ix.**else**x.chrm3, chrm4 = chrm1, chrm2xi.**end if**xii.**if** rand_m1_ < probability of mutation, **then**xiii.mutate chrm3xiv.**end if**xv.**if** rand_m2_ < probability of mutation **then**xvi.mutate chrm4xvi.**end if**xvii.choose 2 best chromosomes from chrm1, chrm2, chrm3, chrm4 & add them to pop (it + 1)xix.it = it+1

**end while**


## 5. Test System and Simulation Setup

This section describes the test system used to evaluate the performance of the proposed strategy, together with details on the EV fleet.

### 5.1. Test System 

To verify the performance of the proposed strategy, an IEEE 33-node, medium-voltage network, connected with low-voltage residential and commercial feeders has been considered, as shown in Figure 4.

The test system supplies residential and commercial customers. Among 32 laterals, 10 are dedicated to commercial customers, and the rest serve residential customers. The line and ad data are obtained from [11]. The daily load profiles for both types of customers are shown in Figure 5. According to the load profiles, the maximum demands for residential and commercial loads are 1.1197 p.u. and 1.122 p.u. which occur at 05:00 PM and 03:00 PM, respectively. The residential peak demand occurs in the evening when people return to their homes and switch on appliances such as air conditioners, lights, TVs and other household requirements [22]. The commercial peak occurs at a time when most commercial buildings use electricity at the same time, often around 03:00 PM [23].

The EV charging activities are assumed at all the 22 residential laterals and 5 commercial laterals. The EV charging aggregators regulate the charging activities in these laterals. Each aggregator is assigned to a lateral and is responsible for achieving the charging satisfaction of the EVs customers of its vicinity.

### 5.2. EV Fleet Specifications

In this study, we have considered 992 EVs with 100% penetration and 496 EVs at 50% penetration performing charging activities in 22 residential and 5 commercial charging platforms. Each charging platform employs level-2 charging standards and the EV charging aggregator governs it. The arrival SoCs are randomly distributed above 20% to maintain an 80% depth of discharge. The specifications, such as EV battery capacity, charger rating, EV penetration and residential and commercial charging events, are given in Table 2.

### 5.3. EVs Mobility Behaviour

The charging behavior is a random phenomenon and it depends on the travel pattern of EV customers. It is very important for the EV charging aggregator to know the mobility pattern of every individual EV so that the charging demands of EV activities can be managed effectively. To this end, the survey results of [27] are used to know about the charging behavior of EVs. The study revealed that more than 50% of EV customers preferred home charging when they returned to their homes in the evening after completing their daily business activities. The second-largest group of EV users are those who use both home and workplace charging, and the workplace charging platform is the choice of only a few percent of EV users. Among the three groups of EVs, 53% of EVs are for home charging, 14% are for workplace charging and the remaining 33% of EVs use both home and workplace charging platforms, as given in Table 2. Based on the findings of the study [27], we have considered the random arrival of EVs for the three groups of EVs as shown in Figure 6.

### 5.4. Electricity Pricing Schemes

To realize the economic benefits of coordinated charging, two dynamic pricing schemes, i.e., (i) ToU pricing and (ii) real-time pricing (RTP), are investigated. ToU prices usually consist of three ToU blocks: off-peak, mid-peak and on-peak blocks, which apply to a specific time of day. Off-peak rates are used when the energy demand is low, whereas mid-peak occurs when the energy demand is moderate. For the maximum energy demand, on-peak rates are applied. ToU pricing can be applied to both residential as well as commercial consumers. The price variations offer consumers the opportunity to adjust their load demand according to their choice. Unlike ToU, RTP is updated mostly every 30 min and offers greater flexibility to consumers to adjust their electricity consumption. RTP is based on real-time electricity generation costs. As peak power plants are more expensive to operate than baseload power plants, electricity rates during peak hours are higher than shoulder and off-peak hours under real-time pricing. In this research work, RTP is assumed as an arbitrary variable. The purpose is to select the most suitable time to start the EV charging activity so that the charging cost is minimized. The ToU [28] and RTP [29] are shown in Figure 7 in units of USD/kWh.

## 6. Results and Discussion

In this section, multiple test scenarios are presented and discussed in detail. The total simulation time is 24 h, and the length of each time step is 30 min. The simulations are performed using MATLAB (R2015a) with an Intel (R) Core i5-5200, a CPU @ 3.40 GHz and 4.0 GB RAM computer specifications. The simulation cases considered in this study to verify the performance of the proposed scheme method are illustrated in Table 3.

### 6.1. Case: A: Uncoordinated EV Charging

Case A is considered as a base case where EV charging activities are not regulated by any control scheme. As soon as the EVs arrive at the charging points, they start charging regardless of the network status and user preferences. The grid performance in terms of power losses, voltage deviations and total power consumption during uncoordinated charging is shown in Figure 8 and summarized in Table 4. Referring to Figure 8a, it is observed that the random charging activities, performed either at homes or at workplaces, increased the network power losses. During the uncoordinated charging activities, the minimum voltage recorded at feeder 18 is 0.8931 p.u. with EV penetration level-I, and it is further reduced to 0.8862 p.u. with penetrating level-II as shown in Figure 8b. Moreover, the total network power consumption is increased for both penetration levels using either platform as shown in Figure 8c. It is recognized that the impact on the network performance is more dominant with EV penetration level-II using the workplace as the charging platform.

Besides this, the economic impact of random EVs charging is shown in Figure 9. The total charging costs with the RTP scheme are USD 1340.0947 and USD 2680.1894 for penetration levels-I and II, respectively. The ToU tariff scheme further increased this cost to USD 1482.0960 and USD 2964.1920 for penetration levels-I and II, respectively. The impact of EV charging on customer satisfaction is summarized in Table 5.

### 6.2. Case: B: Coordinated EV Charging with ToU and RTP Tariff for Waiting Time Minimization

For case B, the coordinated charging operation of EVs with waiting time minimization was carried out as one of the objectives of the second stage. In this case, EV customers do not care about the cost; they want to have the required SoC without undue delay. The technical impacts of this case are illustrated in Figure 10 and summarized in Table 4. It has been found that the network performance is enhanced significantly by reducing the network power losses and increasing the voltage profile of the system for each penetration level. The network power losses are 6266.4944 kW for penetration level-I and 6591.8251 kW for penetration level-II. These losses are reduced by 4.5% and 7.8%, respectively, compared to case A. Similarly, the voltage profile of the system is improved from 0.8931 p.u. to 0.9024 p.u. for penetration level-I and 0.8862 p.u. to 0.9010 p.u. for penetration level-II, respectively, as compared to case A.

In addition to network performance, the impact of coordinated charging on customer satisfaction in terms of early charging and cost is shown in Figure 11. Referring to Figure 11a, the optimization algorithm scheduled the EV’s charging requests so that the connection time closely matches with arrival time, thereby minimizing the waiting time of the EVs participating in the charging process. The average waiting time is recorded as 1.0 and 1.5 h for penetration levels I and II, respectively. This waiting time is higher as compared to case A because in case A, there is no control over the charging activities as soon as the EVs arrive at the charging point; their charging process starts with the violation of network constraints. However, in this case, the EV aggregator agents accommodate the charging demand so that their delay time is minimized without violating the network constraints. Additionally, compared to case A, the charging cost, although not considered as the objective, is reduced to USD 1097.67 and USD 2303 with ToU pricing for penetration levels I and II, respectively, and it is further decreased to USD 1028.46 and USD 1974.67 with the RTP scheme for each penetration level, respectively, as shown in Figure 11b and summarized in Table 5.

### 6.3. Case: C: Coordinated EV Charging with ToU tariff and RTP for Charging Cost Minimization

Besides considering all two objectives of stage-1, this case only deals with charging cost minimization as the objective for stage-2. The technical results for case C are given in Table 4 and depicted in Figure 12. In case C, the power losses are 6249.5636 kW and 6509.6295 kW for penetration levels I and II, respectively. The recorded losses are lower for each penetration level compared to both cases A and B. The voltage quality is improved in this case as the voltage deviation is less as compared to previous cases, summarized in Table 4. 

From a charging satisfaction perspective, EV customers are economically highly satisfied as their charging demand for all penetration levels is managed in time slots when electricity prices are low. For example, the charging cost index with ToU pricing is reduced to USD 618.588 and USD 1388.34 for penetration levels I and II, respectively, as compared to cases A and B, as shown in Figure 13a and Table 5. However, to achieve the minimum charging cost, the EV customers must wait for long time intervals as shown in Figure 13b. The waiting time indices for penetration levels I and II are increased to 7.55% for penetration level-I and 12% for penetration level-II, as summarized in Table 5.

### 6.4. Case: D: Coordinated EV Charging with ToU and RTP Tariff for Both Waiting Time and Charging Cost Minimization

In this case, all the objectives of stages 1 and 2 are considered. The weighting factor among the objectives is uniformly distributed so that a fair analysis can be performed. The technical impacts of this case are presented in Figure 14 and summarized in Table 4. The power loss for each penetration level is the least compared to all the test cases. For example, the power loss for penetration levels I and II are 6191.5129 kW and 6495.0645 kW, respectively; these losses are the least compared to all stated test cases. Moreover, this case avoids the overloading of the network assets and maintains the voltages at each bus within the allowable limit.

In this case, EV customers’ priorities are explicitly determined based on their arrival and departure information. For example, a customer, on arrival at the charging point, specifies a long departure time; this indicates that the customer is more concerned about the cost rather than early charging. A delay in the charging process for such an EV customer gives more satisfaction to them. On the other hand, EV customers with short departure times need early charging at all costs; the satisfaction of these customers is with earlier charging rather than the cost. Keeping in view the explicitly determined priorities of EV customers, this case gives promising results. The charging cost of each penetration level with both ToU and RTP schemes is shown in Figure 15a. The charging cost for case D is lower compared to cases A and B, as summarized in Table 5. However, it is high compared to case C because in case C, all the charging activities are scheduled with a delay so that the minimum charging cost is obtained. However, this does not represent the practical case where few EVs want early charging and others care about cost. The comparatively high charging cost is due to scheduling the charging load of such EVs at the earliest availability of charging facility. For such EV customers, charging satisfaction lies in early charging instead of the charging cost and it is achieved by scheduling them at the earliest as shown in Figure 15b. For those EV customers who are more concerned about the charging cost, their charging requests are scheduled later so that their charging satisfaction in terms of cost is achieved.

To highlight the net effect of the various adopted strategies, ranging from case A to D, a comparison in terms of the overall charging satisfaction index, which is the combination of minimum charging cost and waiting time indices, is presented in Table 6 and illustrated in Figure 16. It can be seen that case D, which considers both the charging cost and the waiting time as the customers’ satisfaction indices, leads to all the other stated cases. Case D gives the highest satisfaction to the customers in terms of cost as well as early charging based on their needs.

## 7. Comparison with the Reported Work

The comparison of the proposed study with the literature in terms of characteristics is presented in Table 7. The proposed study encompassed the charging activities at both platforms, i.e., the home and the workplace, ensured better network performance and satisfied EV customers’ charging objectives in terms of cost and early charging without undue delay. In addition, the management of charging activities by EV charging aggregator agents reduces the search space, as each aggregator agent is responsible for monitoring the charging process at its location and does not have any concerns about neighboring EVs, resulting in better computing performance.

## 8. Conclusions 

In this paper, we have presented a two-stage framework to optimize the performance of the electric grid and maximize the charging satisfaction of EV customers. In the first stage, the DNO receives charging requests from multi-EV aggregator agents and allocates suitable amounts of power to participating agents by minimizing network power losses and voltage deviations. Once power is allocated to EV aggregator agents, the agents then distribute the allocated power among EVs operating in their locality to minimize charging costs and waiting times based on customer requirements. The proposed method is implemented on the IEEE-33 medium voltage system connected with a low-voltage network. Four test cases are presented; in Case-A, random or uncontrolled charging is executed where the highest network power losses and voltage deviation are recorded. Moreover, this case also dissatisfies the customers regarding the minimum charging cost. In Case-B, the coordinated EV charging is implemented by considering all the objectives of stage-1 with only waiting time minimization as the objective of stage-2. In this case, the waiting time is reduced; however, EV customers have to pay more to receive early charging. In Case-C, charging cost minimization is considered as the only objective of stage-2, besides considering all the objectives of stage-1. In this case, the customers care about the cost, and they are not in a hurry about charging their vehicles. The results show that customers are able to achieve minimum charging costs, however with delayed charging. Finally, in Case-D all the objectives of stages 1 and 2 are considered. The simulation results verify that the proposed method outperforms in terms of network performance improvement and achieving the overall charging satisfaction of the EV customers. Future work will focus on the V2G capability of EVs in providing ancillary services to the electricity grid along with the integration of renewable energy sources such as photovoltaics and wind to minimize the dependency on the electric grid.

## Figures and Tables

**Figure 1 sensors-23-02925-f001:**
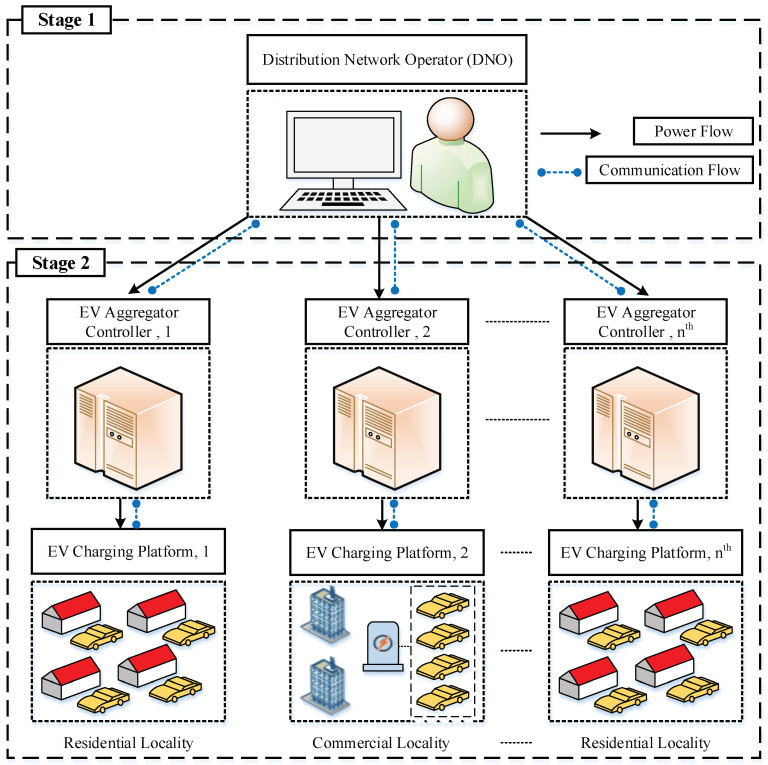
AECM architecture.

**Figure 2 sensors-23-02925-f002:**
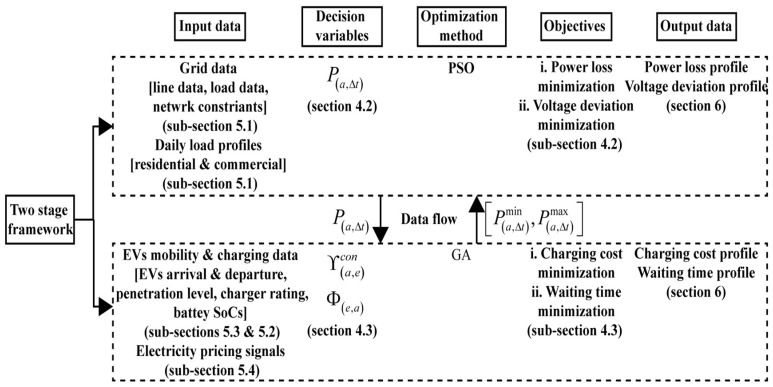
Proposed research design.

**Figure 3 sensors-23-02925-f003:**
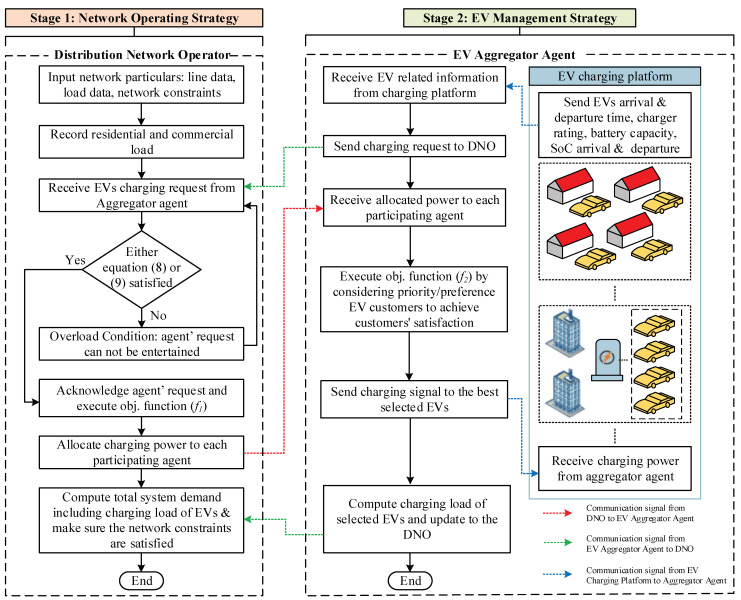
Coordination strategy between two stages.

**Figure 4 sensors-23-02925-f004:**
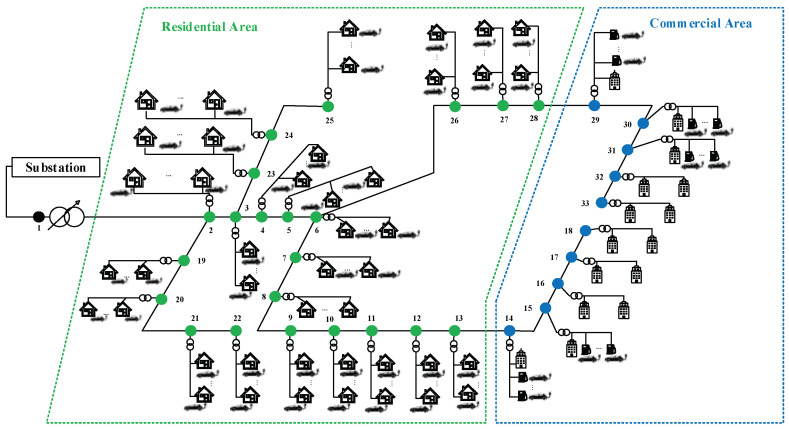
IEEE 33-bus system, connected with low-voltage residential and commercial feeders. (The numbers, 1–33, show the corresponding nodes).

**Figure 5 sensors-23-02925-f005:**
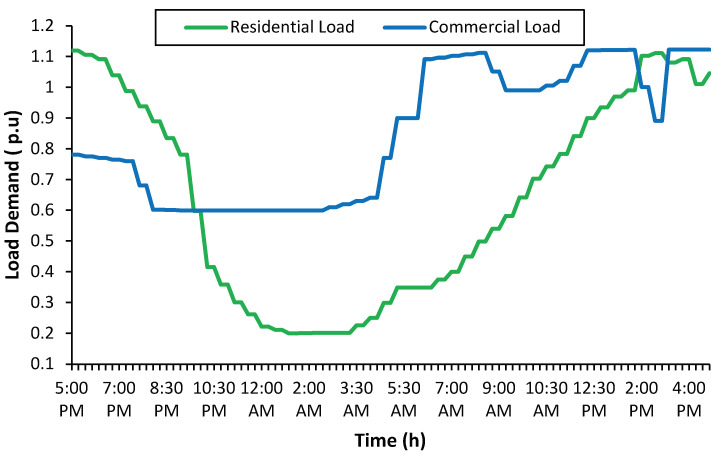
Daily residential and commercial demand profiles [24].

**Figure 6 sensors-23-02925-f006:**
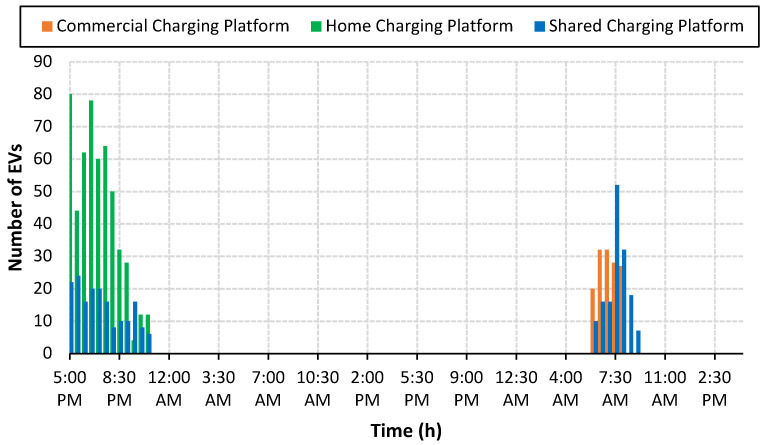
EV arrival at different charging platforms.

**Figure 7 sensors-23-02925-f007:**
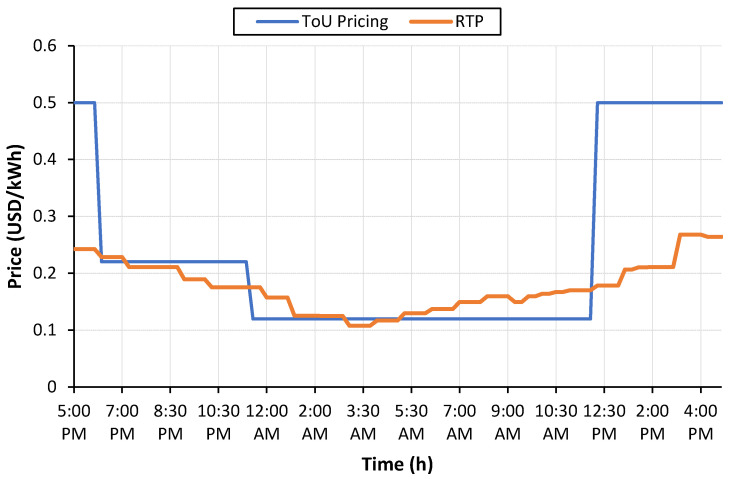
ToU and RTP signals.

**Figure 8 sensors-23-02925-f008:**
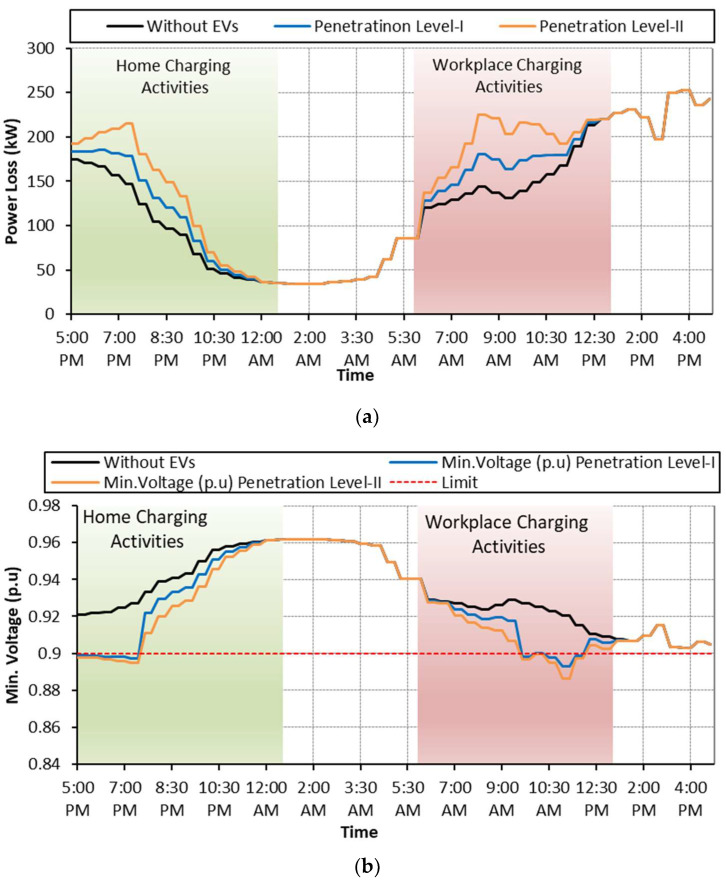
**Case A:** (**a**) Network power loss; (**b**) voltage deviation; (**c**) network power consumption.

**Figure 9 sensors-23-02925-f009:**
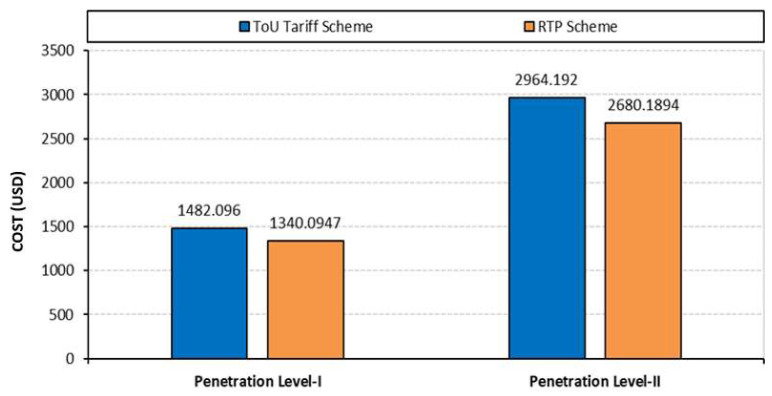
Impact of various tariff schemes on the charging cost.

**Figure 10 sensors-23-02925-f010:**
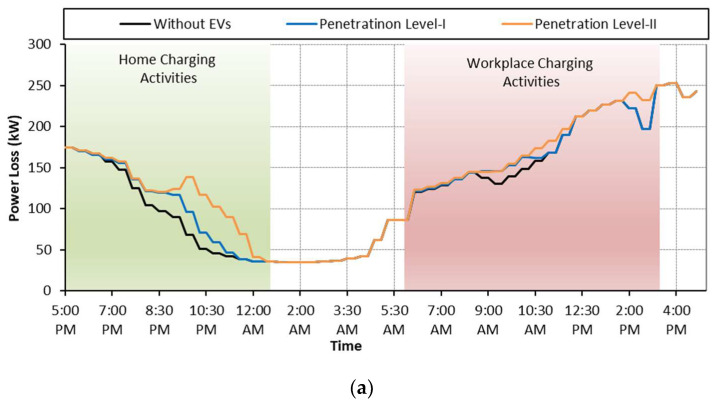
**Case B:** (**a**) Network power loss; (**b**) voltage deviation; (**c**) network power consumption.

**Figure 11 sensors-23-02925-f011:**
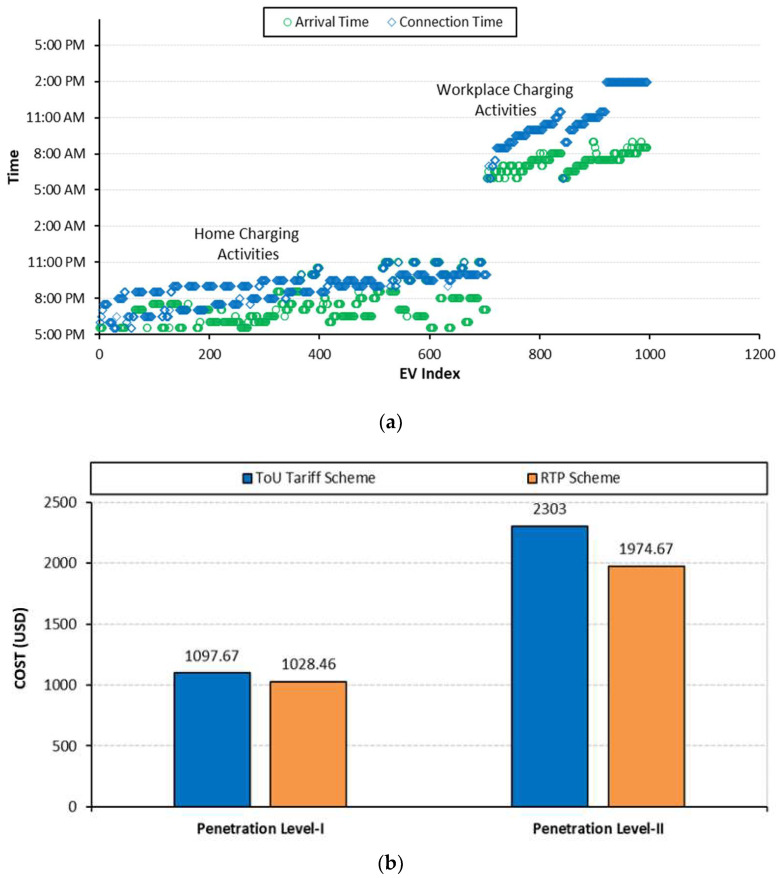
**Case B:** (**a**) Waiting time; (**b**) charging cost.

**Figure 12 sensors-23-02925-f012:**
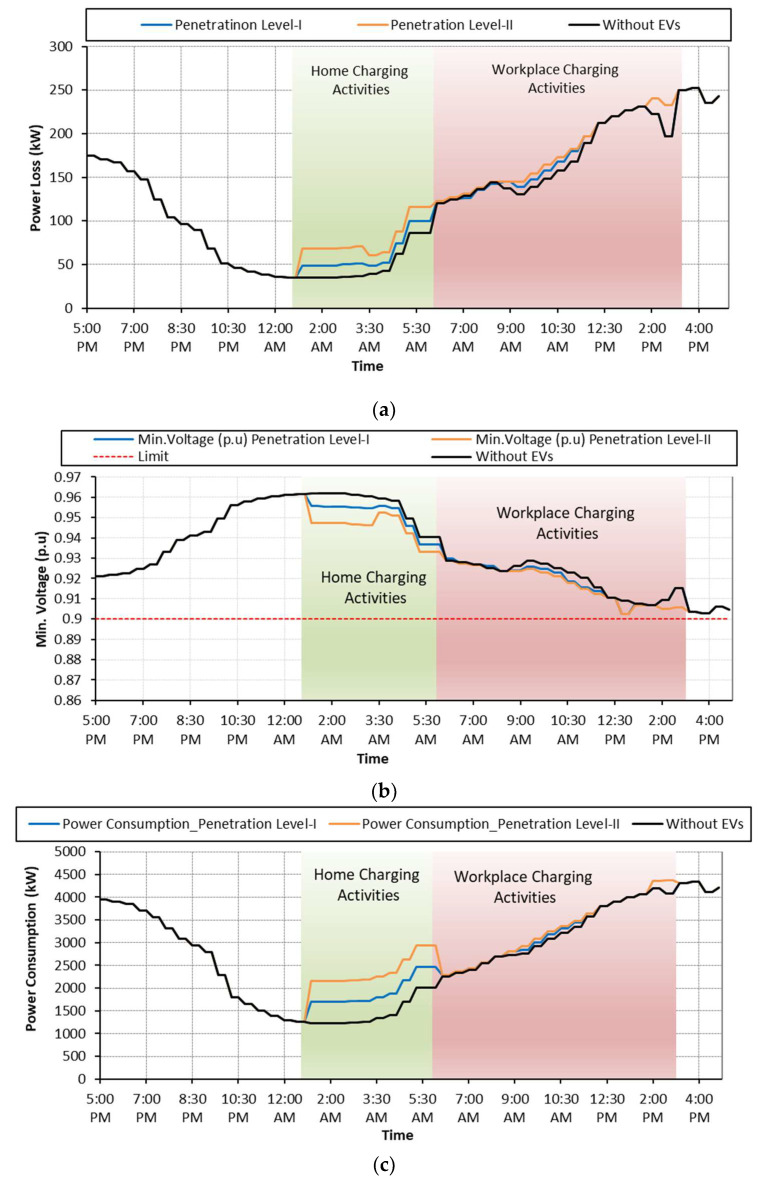
**Case C** (**a**) Network power loss; (**b**) voltage deviation; (**c**) network power consumption.

**Figure 13 sensors-23-02925-f013:**
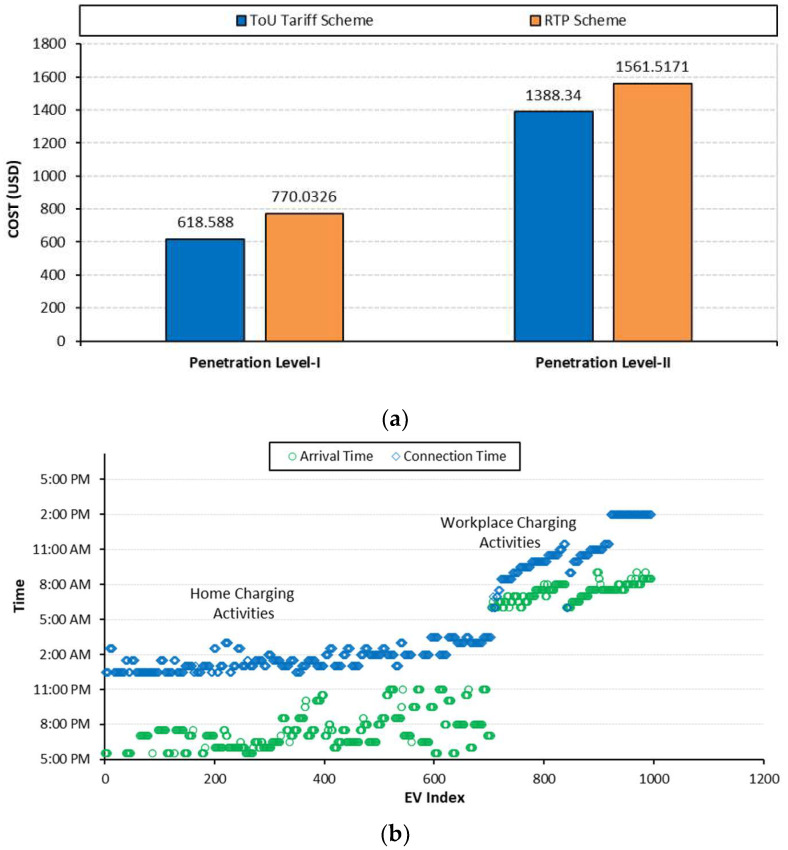
**Case C:** (**a**) Charging cost; (**b**) waiting time.

**Figure 14 sensors-23-02925-f014:**
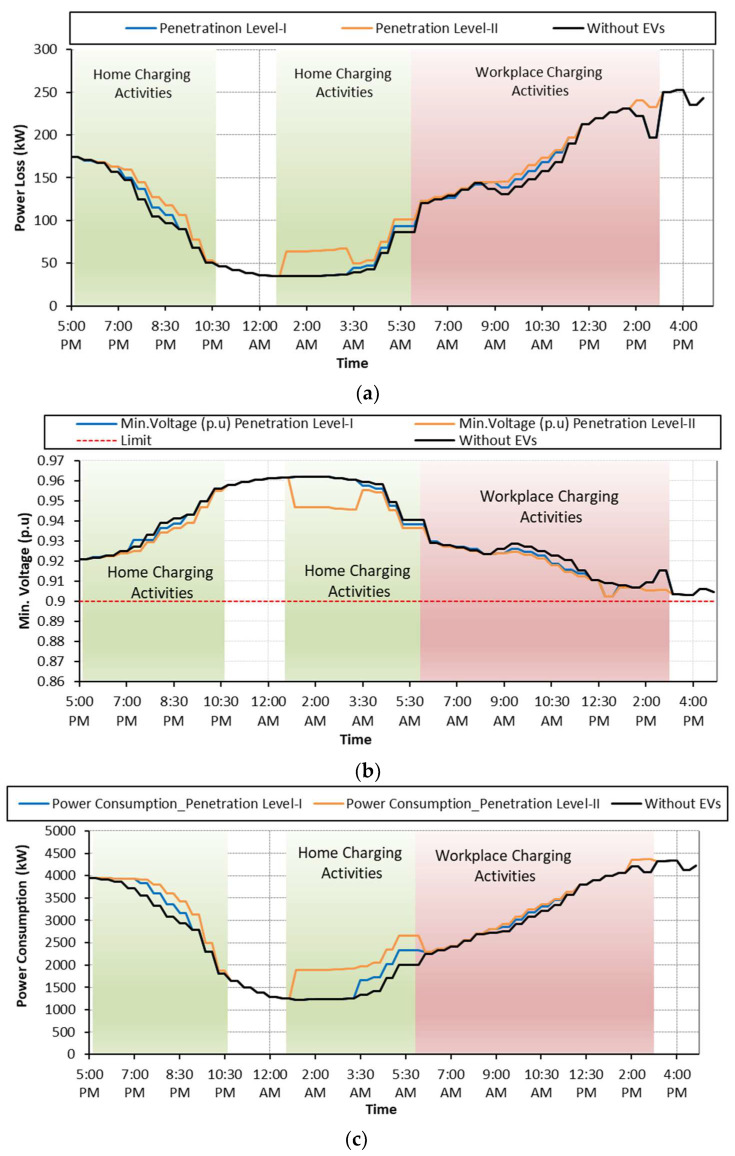
**Case D:** (**a**) Network power loss; (**b**) voltage deviation; (**c**) network power consumption.

**Figure 15 sensors-23-02925-f015:**
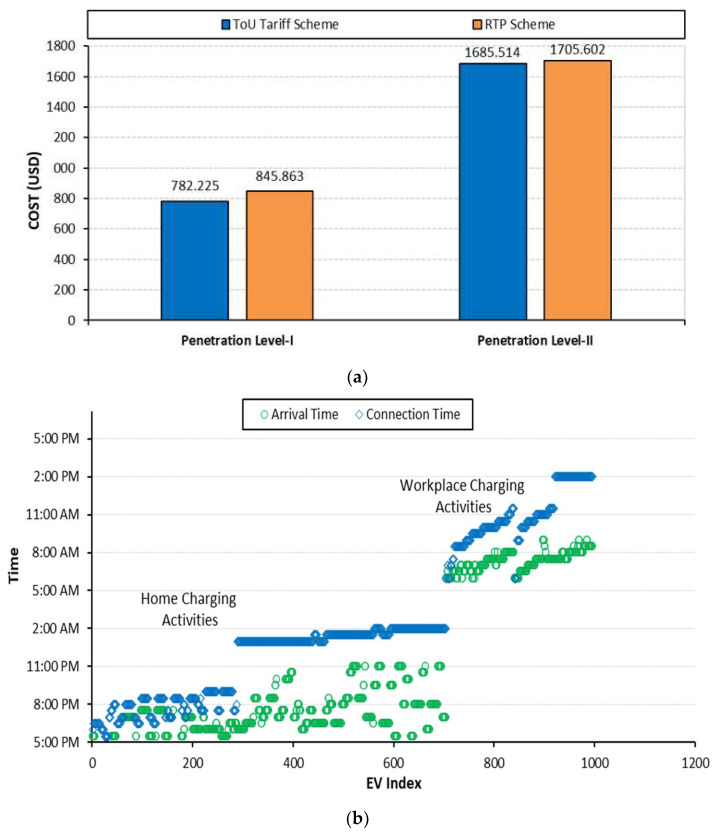
**Case D:** (**a**) Waiting time; (**b**) charging cost.

**Figure 16 sensors-23-02925-f016:**
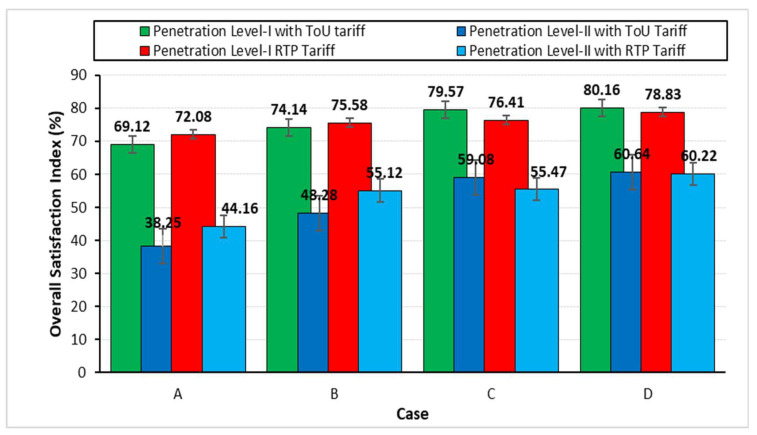
Comparison of overall charging satisfaction index.

**Table 1 sensors-23-02925-t001:** Summary of related work.

Ref.	EV Charging Control Architecture	Research Objectives	Optimization Method	Charging Platform	Pricing Scheme
[11]	Centralized	Power loss minimization	Binary PSO	Residential	Not applicable
[12]	Centralized	Power loss minimization	Binary EP	Residential	Not applicable
[13]	Centralized	Cost and system stress minimization	Binary PSO	Residential	ToU
[14]	Hierarchal	Peak shaving and valley filling	Water-filling algorithm	Residential	Not applicable
[15]	Centralized	Power loss and charging cost minimization	Binary PSO and analytical hierarchy process	Residential	ToU
[16]	Hierarchal	Peak shaving and valley filling	Heuristic	Residential	Not applicable
[17]	Centralized	Peak power, power losses and cost minimization	PSO	Residential	RTP
[18]	Hierarchal	Electricity distribution and charging schedule optimization	PSO	Residential	ToU
Proposed	Hierarchically centralized	Power loss, voltage deviation, charging cost and waiting time minimization	PSO, GA	Residential, Commercial	ToU and RTP

**Table 2 sensors-23-02925-t002:** EVs Fleet Specifications.

Battery Capacity [13,25,26]	Charger Rating [13]	Number of EVs		Residential Charging Fleet [27]	Workplace Charging Fleet [27]	Common Charging Fleet [27]
		Penetration I	Penetration II			
10.5 kWh	3.3 kW	176	352	53%	14%	33%
19.2 kWh	6.6 kW	164	328			
20.7 kWh	7.2 kW	156	312			

**Table 3 sensors-23-02925-t003:** Simulation cases.

Case No.	Description	Tariff Scheme		Waiting Time Minimization	Charging Cost Minimization
		ToU	RTP		
A	Uncoordinated EV Charging	√	√	√	√
B	Coordinated EV charging with ToU and RTP tariff for waiting time minimization	√	√	√	O
C	Coordinated EV Charging with ToU and RTP tariff for charging cost minimization	√	√	O	√
D	Coordinated EV Charging with ToU and RTP tariff for both waiting time and charging cost minimization	√	√	√	√

O = No, √ = Yes.

**Table 4 sensors-23-02925-t004:** Impact of EV charging on the network performance for all four cases.

Performance Parameters	EV Penetration Levels	Cases			
		A	B	C	D
Total Power Loss (kW)	Level-I	6562.9858	6266.4944	6249.5636	6191.5129
	Level-II	7150.0197	6591.8251	6509.6295	6495.0645
Power Loss Index PLI (%)	Level-I	7.6933	3.3259	3.0640	2.1551
	Level-II	15.2719	8.0971	6.9367	6.7280
Min. Voltage (p.u)	Level-I	0.8931	0.9024	0.9320	0.9028
	Level-II	0.8862	0.9010	0.9301	0.9023
Voltage Deviation Index VDI (%)	Level-I	7.9199	7.5421	7.5430	7.5633
	Level-II	8.6309	7.9238	7.8584	7.8787

**Table 5 sensors-23-02925-t005:** Impact of EV charging on customer charging satisfaction for all four cases.

Performance Parameters		EV Penetration Levels	Cases			
			A	B	C	D
Total Charging Cost (USD)	ToU	Level-I	1482.09	1097.67	618.588	782.22
	RTP	Level-I	1340.09	1028.46	770.0326	845.86
	ToU	Level-II	2964.19	2303.00	1388.34	1685.5
	RTP	Level-II	2680.18	1974.67	1561.5171	1705.60
Charging Cost Index CCI (%)	ToU	Level-I	30.88	22.86	12.88	16.29
	RTP	Level-I	27.92	21.42	16.04	17.62
	ToU	Level-II	61.75	47.97	28.92	35.11
	RTP	Level-II	55.84	41.13	32.53	35.53
Average Waiting Time (Hr.)		Level-I	0	1	2.5	1.5
		Level-II	0	1.5	3	2
Waiting Time Index WTI (%)		Level-I	0	3	7.55	3.55
		Level-II	0	3.75	12	4.25

**Table 6 sensors-23-02925-t006:** Overall charging satisfaction index for each case.

Case No.	EV Penetration Levels	Charging Cost Index CCI(%)		Waiting Time Index WTI(%)	Overall Charging Satisfaction Index ϑ(%)=100%−[CCI(%)+WTI(%)]	
		ToU	RTP		ToU	RTP
A	Level-I	30.88	27.92	0	69.12	72.08
	Level-II	61.75	55.84	0	38.25	44.16
B	Level-I	22.86	21.42	3	74.14	75.58
	Level-II	47.97	41.13	3.75	48.28	55.12
C	Level-I	12.88	16.04	7.55	79.57	76.41
	Level-II	28.92	32.53	12	59.08	55.47
D	Level-I	16.29	17.62	3.55	80.16	78.83
	Level-II	35.11	35.53	4.25	60.64	60.22

**Table 7 sensors-23-02925-t007:** Comparison of proposed work with the reported work.

Ref.	Objectives	Techniques Applied	Different Charging Platforms		Customer Satisfaction		Multi-Agent-Based Control of Charging Activities
			Home	Workplace	Cost	Early Charging	
[11]	Power loss minimization	BPSO	√	O	O	O	O
[12]	Power loss minimization	BEP	√	O	O	O	O
[15]	Power loss minimization, charging cost minimization	BPSO and AHP	√	O	√	O	O
[13]	Cost and system stress minimization	BPSO and BGWO	√	O	√	O	O
[14]	Load variance minimization	WFA	√	O	O	O	√
[16]	Peak shaving and valley filling	Heuristic approach	√	O	O	O	√
Proposed work	Minimization of power loss, voltage deviation, charging cost and waiting time	PSO and GA	√	√	√	√	√

O = No, √ = Yes.

## Data Availability

The data presented in this study are available upon request. The code for this paper is uploaded on a GitHub https://github.com/ahsanrazakhan/EV-Charging-Scheme.git.

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
