# Peer review of "A Two-Stage Multi-Agent EV Charging Coordination Scheme for Maximizing Grid Performance and Customer Satisfaction"

_sensors, 2023, doi:10.3390/s23062925_

Round 1

Reviewer 1 Report

This paper proposes a two-stage multi-agent-based scheme to realize the coordinated charging scheduling of EVs. Finally, this paper successfully improves the network performance and customers' charging satisfaction. Following are some of my questions and comments in this regards:

1. There are some format problems in the paper. For example, on page#6 line 207, there is an extra semicolon; On page#7, Fig.2, the formula part is too vague to recognize; There is an unreasonable line break on page#11 line 319.

2. I believe when analyzing Fig.4, the authors should not only point out the time when the peak load demand occurs, but also explain the reason why the peak load demand occurs. In addition, the residential load in Fig.4 has an obvious peak at about 3 o'clock. Is it reasonable? Because the peak load in the residential area should occur in the off-duty peak period, while 3:00 p.m. is not the off-duty peak period.

3. On page#4 line 128, what is the basis for the classification of charging levels in this paper?

4. On page#4 line 153, how do EV aggregators obtain the desired departure SoC and departure time? The charging station can only know the users' arrival time and initial SoC, while users' departure SoC and departure time can only be obtained after the charging process is completed.

5. In this paper, the real-time pricing scheme is adopted, but the authors don’t explain how the charging price is set in different time periods. However, the charging price in different time steps will directly affect the charging cost of users, so this part should be supplemented in Section 3.3.

Author Response

Please see the attachmet.

Reviewer 2 Report

-Author should discuss more refernces in introduction section as below

"Assessing the charging load of battery electric bus fleet for different types of charging infrastructure"

"Effect of local grid refinement on performance of scale-resolving models for simulation of complex external flows"

"Partial grid false data injection attacks against state estimation"

"Job scheduling problem in fog-cloud-based environment using reinforced social spider optimization"

-Author should add border in all tables.

-future work should more focused and clearly mention at the end of conclusion.

-please give a proofread check to the paper.

Reviewer 3 Report

The article is interesting but some points are not clear.

The open problems are not evident in the introduction, also in the introduction and throughout the text, the authors do not show concretely which research problems were solved.

A section with related works is imperative, it is an article for a journal and not a conference, so this section making it clear to the reader which literature was researched and compared in relation to this work is imperative.

Sections 2 and 3

I could not understand the values and logic used, what were the calculation criteria? what parameters were considered? what's the comparison? evidence is needed. There is no code snippet throughout the text for the reader to understand.

It makes sections 4 and 5 confusing and of little understanding, so this reviewer doubts the real scientific contribution of the work developed. For these sections, I suggest that you make code snippets available, perhaps in some repository so that the reader can reproduce them in other scenarios, thus proving the scientific contributions.

I confess that I spent hours trying to understand the figures' and graphs' data and values without success.

Note that the author's conclusion is unclear about the news obtained, the problems solved, or the scientific contributions, and there are not even definitions for future work.

Round 2

Reviewer 1 Report

thank you for your reply

Author Response

Thank you so much for providing your critical feedback to improve our work. We believe your concerns were valid, and after addressing those issues, the manuscript is in much-improved shape.  

Further, there we no specific comments in the second round, apart from English, so we tried our best to improve the language.  

Reviewer 3 Report

Dear authors, you have made some improvements to your work, but it is not clear which metrics were used to compare the state of the art.

This article by the same publisher involves scenarios and I believe it can be useful in an integration context, in addition to contributing https://doi.org/10.3390/s20102849

I am also not satisfied enough with the results you present in section 5, I could not understand how you arrived at the values presented in the graphs.

I suggest including a step-by-step, describing the variables, parameters, and data that were considered.

Use, if necessary, a repository like GitHub to make used datasets available, so readers can reproduce the simulations in other scenarios.

Include a list of abbreviations

Include a list of math symbols

Several references have not yet been adjusted, including DOI or ISSN.

The work is interesting but needs adjustments, I hope I have contributed to the improvement of your work, looking forward to receiving an updated version.

Round 3

Reviewer 3 Report

For the final version, I suggest revising the text, and formatting.